# A Low-Cost AI Buoy System for Monitoring Water Quality at Offshore Aquaculture Cages

**DOI:** 10.3390/s22114078

**Published:** 2022-05-27

**Authors:** Hoang-Yang Lu, Chih-Yung Cheng, Shyi-Chyi Cheng, Yu-Hao Cheng, Wen-Chen Lo, Wei-Lin Jiang, Fan-Hua Nan, Shun-Hsyung Chang, Naomi A. Ubina

**Affiliations:** 1Department of Electrical Engineering, National Taiwan Ocean University, Keelung 202301, Taiwan; hylu@mail.ntou.edu.tw (H.-Y.L.); 10953054@mail.ntou.edu.tw (Y.-H.C.); 00653005@mail.ntou.edu.tw (W.-C.L.); caota985107@gmail.com (W.-L.J.); 2Department of Computer Science and Engineering, National Taiwan Ocean University, Keelung 202301, Taiwan; csc@mail.ntou.edu.tw (S.-C.C.); naomi.ubina@gmail.com (N.A.U.); 3Department of Aquaculture, National Taiwan Ocean University, Keelung 202301, Taiwan; fhnan@mail.ntou.edu.tw; 4Department of Microelectronics Engineering, National Kaohsiung University of Science and and Technology, Kaohsiung 811213, Taiwan; shchang@nkust.edu.tw

**Keywords:** artificial intelligence, offshore aquaculture, wireless communications, machine learning, water quality

## Abstract

The ocean resources have been rapidly depleted in the recent decade, and the complementary role of aquaculture to food security has become more critical than ever before. Water quality is one of the key factors in determining the success of aquaculture and real-time water quality monitoring is an important process for aquaculture. This paper proposes a low-cost and easy-to-build artificial intelligence (AI) buoy system that autonomously measures the related water quality data and instantly forwards them via wireless channels to the shore server. Furthermore, the data provide aquaculture staff with real-time water quality information and also assists server-side AI programs in implementing machine learning techniques to further provide short-term water quality predictions. In particular, we aim to provide a low-cost design by combining simple electronic devices and server-side AI programs for the proposed buoy system to measure water velocity. As a result, the cost for the practical implementation is approximately USD 2015 only to facilitate the proposed AI buoy system to measure the real-time data of dissolved oxygen, salinity, water temperature, and velocity. In addition, the AI buoy system also offers short-term estimations of water temperature and velocity, with mean square errors of 0.021 °C and 0.92 cm/s, respectively. Furthermore, we replaced the use of expensive current meters with a flow sensor tube of only USD 100 to measure water velocity.

## 1. Introduction

In the past decade, global warming, climate change, ocean pollution, overfishing, etc., have caused marine catches to be rapidly attenuated [1,2,3,4,5]. This phenomenon reflects the problems mentioned earlier regarding the rapid depletion of marine resources. In the long run, the depletion of marine catches will inevitably lead to food shortages and even the possibility of endangering the survivability of human beings. With this threat to food security, many researchers in different fields have been profoundly motivated and challenged to focus and seek feasible countermeasures and technologies [5,6,7] to stabilize and increase food production. Their studies and physical experiments revealed that offshore cage cultures in aquaculture have a significant role in compensating for the depletion of marine catches [8,9,10]. Moreover, to further enhance aquaculture production, many governments and groups have been devoted to developing new technologies and devices for aquaculture in recent years [11,12,13,14,15,16,17,18]. For ease of reading, we briefly list the technologies of [11,12,13,14,15,16,17,18] in Table 1.

Various studies and literature works show water quality as one of the key factors affecting the quality and quantity of aquaculture [19,20,21,22,23,24] production. In addition, real-time water quality monitoring is the first step to facilitating suitable and conducive aquaculture [25] environment. The essential parameters of water quality for successful aquaculture production includes water temperature, dissolved oxygen (DO), salinity, turbidity, pH, and conductivity [26,27]. Oxygen is essential for the respiration of all biological creatures, including various marine and estuarine organisms. Moreover, the feed intake of fishes in high DO environments is significantly higher than those in low DO [24]. Hence, in the monitoring, DO is an indicator of water quality in coastal areas. In addition, DO levels are highly dependent on physio-chemical parameters, including temperature and salinity, necessitating the monitoring of these water quality parameters [23]. For example, the low water temperature may reduce the amount of food intake for some farmed fish species [20,21]. Concerning the feeding requirements for offshore cage culture, water velocity and direction are also necessary [28] since faster water velocity can drive floating fish food out of the cages [13,22]. Hence, without real-time monitoring of water quality such as low water temperature and rapid water velocity [19], farming efforts may lead to food waste, increasing production costs, and even water pollution. With all the given premises for successful aquaculture operations, water quality monitoring in real time can contribute to the success of aquaculture farming [17,18]. However, traditional methods and devices are challenging to install or operate in a typical or standard aquaculture farm environment, and they usually come with higher costs. Therefore, integrating real-time water quality monitoring that is feasible and with reduced cost in its implementation is a vital and compelling design consideration.

Recently, the focus of investigations concerning wireless communications and the Internet of Things (IoT) are aiming to improve in real-time in terms of monitoring of Zigbee, a technology of sensor network that collects water quality measurements of fish cages and then sends them to the terrestrial server via the 3G phone system [12]. The authors in [17] utilized Raspberry Pi, a WiFi-based microprocessor platform, to construct a wireless sensor network to monitor water quality of the aquaculture site. On the other hand, another popular microprocessor platform, Arduino, has also been used to build an IoT network to measure water quality [18]. Similarly, a WiFi communication-based IoT system was also proposed in [29] to facilitate real-time water quality monitoring. However, unlike the work of [17], it can concurrently measure DO together with ammonia, pH, temperature, salinity, nitrates, and carbonates. Furthermore, in [30], the authors used a hybrid wireless-wired approach to design a practical underwater sensor network for offshore cages, which reaches up to 30-meter depth and mitigates wire breakage problems common to offshore cages. Parra et al. proposed a low-cost sensor network to monitor fish behavior and water quality in aquaculture tanks during the feeding process [31]. Their system can prevent unnecessary information from being sent from the node to the database, thus, reducing power consumption. The various research results mentioned earlier pointed out the benefits of monitoring water quality. Although the benefits are promising, the distance of the cages from the shore measures several kilometers, which challenges, restricts, and limits the advantages of these technologies.

The measurement of water velocity, as mentioned above, is an essential consideration for offshore aquaculture and can be achieved by deploying precise but expensive flow meters. However, these high-cost flow meters might not be affordable for regular fish farmers. Therefore, many researchers were motivated to investigate and design low-cost and affordable technologies for water velocity measurement to help ease the burden for farmers in acquiring such innovations. In the work of Marchant et al. [32], the authors built an electronic accelerometer in a spherical ball and then nailed the ball to the seabed. With this set-up, the action of the current flow on the ball causes angle deviation of the sphere, which also renders the accelerometer to generate the corresponding angle deflection data to be stored in the memory card of the ball. After recording the angle data for a prescribed period, the spherical ball is taken back to the shore for memory-card data retrieval and to compute the corresponding sea water velocity. On the other hand, Beddows et al. [33] used the Arduino platform to build a low-cost logger, which can measure and record the water velocity in a harsh water environment for a longer duration. However, the methods in [32,33] cannot provide real-time water velocity information, limiting their capabilities when implemented in the offshore aquaculture environment.

Recently, artificial intelligence (AI) technologies have been successfully applied in various fields, such as aquaculture [28], Internet of Things (IoT) [34], green communications [35], unmanned aerial vehicles (UAV) [36], and traffic control [37]. In the recent decade, Aquaculture 4.0 has also become the world trend [15,38], which provides the benefits of intelligent automation. The successful applications of AI and the world trend of Aquaculture 4.0 represent a big motivation in developing AI systems to improve the performance of offshore aquaculture. In this paper, the author proposes a low-cost buoy system with AI integration [34,39] to monitor the water quality of offshore cages. The paper’s primary purpose is to develop a buoy system that uses AI technologies to automate water quality measurement and with added prediction capability using short-term data on water temperature and velocity. In addition, the authors carefully considered a low cost and easy to build or deploy buoy system as one of the considerations in its design and architecture. Furthermore, the monitoring results for water velocity, direction, temperature, and other seawater parameters are more complex due to weather conditions offshore. These complex results make it more difficult to predict seawater parameter values. AI is a promising approach to deal with these complex problems. Its strength lies in not knowing the relation between seawater parameters and the kind of their combination causing the complex result due to weather conditions. AI systems are programmed to use external data to learn. It is flexible enough to adapt the connections of the models and then use the generated knowledge to achieve specific goals which makes it a very popular data-driven approach. Accordingly, in training the AI models for the proposed buoy systems, we collected long-term data at Haikou Port, Pingtung, Taiwan, with 734,000 water temperature and 36,237 velocity data. After the AI model training, the proposed buoy system now behaves similar to an expert in predicting short-term water temperature and velocity.

Aside from the prediction capability of the proposed AI buoy system, its main modules contain a sensor measurement mechanism, wireless communication module, power and control module, and three server-side AI programs. Three procedures come with the proposed buoy system. First, are the RS-485 based sensors [40], including the flow sensor tube, which measures the corresponding water quality data specifically dissolved oxygen (DO), salinity, and the accelerometer’s deflection angles. The second is a buoy that uses Long Range (LoRa) remote modules [16] to transmit the measurement data back to the shore server for storage. As a result, the data provides aquaculture staff with real-time water quality information of offshore cages. Lastly, the server-side uses data to train AI programs, which offer short-term predictions on water temperature and velocity information. With the help of the prediction information, aquaculture staff can intelligently decide the amount of bait to feed fish. This approach can enhance the production performance of offshore aquaculture, which will eventually save bait cost, and reduce ocean pollution. The contributions of the this proposed AI buoy system are as follows:The proposed AI buoy system is designed and implemented to achieve a low-cost and easy-to-build architecture that deals with the difficulty of installation in the water environment.RS-485 with an industry interface standard is integrated into the buoy to enhance the stability of sensor measurement. In addition, to adapt to the dynamics of the interface standard, the proposed buoy allows the aquaculture staff to switch to different sensors for various water quality parameter monitoring.Integrating LoRa for the wireless communications mechanism requires low power consumption for the proposed AI buoy operation in transmitting the water quality measurement data, considering it is several kilometers away from the shore server.The measurement data stored at the shore server are utilized for the machine learning algorithm training using the server-side AI programs. The training results provide AI models for intelligent water quality prediction on water temperature and velocity. In addition, the data measured by the flow sensor tube are also utilized to assist the AI regression in estimating water velocity, thereby achieving low-cost water flow meter design and implementation.

The remainder of the paper is organized as follows. The architecture of the proposed buoy system and its operation flow are presented in Section 2. Section 3 discusses the hardware modules of the proposed system. Server-end AI programs deployed at the shore server are detailed in Section 4. Section 5 discusses the implementation results. Finally, conclusions are drawn in Section 6.

## 2. Architecture and Operation Flow

In this section, we first introduce the architecture of the proposed buoy system and then illustrate its operation flow. Figure 1 and Figure 2 show its architecture and implementation results, respectively.

### 2.1. Architecture

As shown in Figure 1, the hardware architecture of the proposed AI buoy is mainly composed of a solar panel, a control box, two lifebuoys, a steel skeleton, sensors, and a water flow sensing tube. In addition, three server-side AI programs are also built at the shore server of the system. The components are further explained in the following items:**Solar panel:** The solar panel converts the irradiated energy of the sun into electrical energy and then stores the energy in the battery, thereby functioning as a power source for the offshore buoy.**Waterproof control box:** The control box was designed to provide space for the kernel devices of the offshore buoy. The devices included are the Arduino which controls the entire function of the buoy, and the LoRa module, which is responsible for wireless communication transmissions.**Lifebuoys:** The two lifebuoys provide the needed buoyancy for the offshore buoy to float on the water surface.**Steel skeleton:** The steel skeleton combines the control box, lifebuoys, and other associated items as a buoy.**Sensors:** Measure water quality data such as temperature, DO, and salinity.**Water flow sensing tube:** An electronic accelerometer is installed to measure water velocity and direction using the flow tube. In addition, the flow tube is hung under the steel skeleton.**Server-side AI programs:** Three AI programs were deployed at the shore server to predict water temperature within the eight-hour duration, for water velocity within the four-hour duration.

Since our primary goals are to facilitate low-cost and easy-to-build features, we avoided complicated structures and time-consuming construction methods in the whole design of the buoy system. To make these possible, feasible and inexpensive materials are adopted to achieve the low-cost goal.

### 2.2. Operation Flow

The operation flow of the proposed buoy system can be divided into two parts: data measurement and storage and water quality prediction. For the data measurement and storage, the buoy placed on the pond or offshore automatically activates at a designated time. Then, the Arduino chip inside the buoy will command the sensors to measure water quality and reads the time data through the Global Position System (GPS) module. Finally, the data (water quality and time) are merged and sent back to the shore server for storage via the LoRa device. Once the requirements are completed, the buoy will return to sleep mode. Meanwhile, for the water quality prediction, the AI programs at the server use the data stored to estimate the current water velocity and predict the changes in the water temperature and velocity over the next several hours as designated. The prediction information of these AI programs are accessed using a mobile application (APP) to provide users with information regarding the current status and future water quality trends. Further, to better understand these concepts, we provided visuals on the operation flow, data measurement and storage in Figure 3.

## 3. Hardware Modules

The key hardware components of the proposed buoy system are the control box, solar panel, and sensors. Likewise, an Arduino chip, a LoRa remote module, a solar controller, a Lithium-ion (Li) battery, and a GPS module were integrated in the control box. The details of their functionalities are discussed below.

### 3.1. Control Box

The buoy needs to be deployed in the water for an extended period of time. Therefore, using an acrylic board as a material to build the control box will achieve the waterproof requirement of the buoy. Figure 4 shows the control box and its internal modules. The following are the internal modules of the control box: 

(1)
*Arduino Chip*


Arduino is a popular micro-control chip owing to its low cost and ease of use. In particular, its flexibility enables different hardware modules and application libraries to be directly integrated, which helps the more straightforward implementation of the system requirements. Moreover, the compatibility of several Arduino platforms for different application environments and conditions make it a very convenient platform for practical implementation. Inside the control box of the proposed buoy, Arduino Mega 2560 manages the control and communication units. The platform of Arduino Mega 2560 has 54 input/output pins and is connected with a 16 MHz crystal oscillator. Furthermore, the platform is equipped with a bootloader mechanism that enables a program to be directly downloaded via the USB interface. For the Arduino platform’s electrical power source, a Lithium-ion battery (demonstrated below) with DC 12 Volts (V) is supplied. The input voltage of Arduino Mega 2560 can range from DC 7 to 12 V.

The Arduino platform’s primary function is to control the proposed buoy. Its control work can be divided into two phases: active and sleeping. The Arduino Mega 2560 first instructs the sensors to measure the water quality, such as temperature and velocity, during the active phase. Next, it collects the measurement data and sends them to the shore server using the LoRa wireless communication device. After that, the Arduino will wait for a short time to receive the acknowledgment from the server; this mechanism confirms if the data transmission is successful. After successful communication, the Arduino Mega 2560 now switches the offshore buoy into the sleeping phase for power saving. Later, based on a schedule, it will again wake up the buoy for another data collection round or another active phase cycle. Based on a practical demand, the period considered is 30-min for the proposed buoy system.

(2)
*LoRa Remote Module*


LoRa is a popular wireless network widely used in the Internet of Things (IoT) [16]. According to the official sheet of LoRa, its longest transmission distance is 15 km and with a maximum data rate of 300 kbps. The offshore cage and the shore server distance are several kilometers away in an aquaculture farm envirnment. Furthermore, the size of the measurement data for water quality is less than 1 kbits, making LoRa a feasible device for the wireless communication network of the proposed buoy system. In addition, some of its advantages include low power consumption, low cost, and no communication fees or charges. These are the reasons for adopting LoRa as the communication device for the proposed buoy using S76S and S76G of Acsip.

(3)
*Lithium-ion Battery and Solar Controller*


In the proposed buoy, we use a Lithium-ion battery, DP-1206A of Doublepow, to supply the electrical power of the offshore buoy deployed at the offshore cage. The battery’s capacity is 21AH, and its operating voltage ranges from 8.5–12.6 V with a charging current range of 1–5 amperes (A). To ensure that the battery can provide the daily power demand, a solar panel, D-30 of Solar World, is utilized and is shown in Figure 1 and Figure 2. The battery is maximally configured in terms of output power, output voltage, and output current of 30 watts, 18.18 V, and 1.65 A, respectively, for the battery charging requirements. In addition, a solar controller, JL-30A of Solar World, is also utilized to connect the Li battery and the solar panel to protect the battery from overcurrent and to provide a stable 5 V operating voltage needed for the Arduino platform.

(4)
*GPS Module*


The GPS signal contains the time information to provide the time stamp for the water quality measurement data. The timestamp provides the exact time and day when the water quality data were measured. A NEO-7M UBLOX GPS module was deployed to receive the GPS signal. This GPS module has low power consumption, high sensitivity, short sampling and receiving time, low cost, and can be connected to DC 3.3/5 V. Furthermore, its SMA antenna interface can be connected to a variety of antennas. We used an antenna with 7 dBi gain to connect to the GPS module, which is placed on the outside of the control box to enhance the receiving performance of the GPS module.

### 3.2. Solar Panel

The goal of the solar panel is to convert and charge the sun’s irradiated energy to the battery to enable the proposed buoy system to perform its daily duties. The detailed electrical specification for the proposed buoy system is in Section 3.1.

### 3.3. Sensors

In this paper, the water quality parameters measured by the proposed buoy system are water temperature, water velocity and flow direction, DO, and salinity. We used the EPK-P1FDo-AL DO and EPK-I1SA-SL salinity sensors, which are the products of eKoPro and are capable of measuring water temperature. To position the two sensors under the water surface, we suspended the sensors under the bottom of the steel skeleton of the proposed offshore buoy. Furthermore, to ensure that the sensors are stable and have a universal interface standard for the transmission of measurements, we adopted RS-485 [40] industry interface. According to RS-485’s protocol, its stable transmission distance can reach 1200 m and its maximum number of modules connected in series to the same logger is 32. With this capability, we can deploy a logger instead of multiple loggers, which meets our goal of cost-saving while reducing the space requirements of the buoy. We considered RS-485 for its flexibility to adapt to future expansion using added measurement parameters.

On the other hand, a water flow sensing tube is suspended under the steel skeleton of the proposed buoy and its interior contains an Arduino chip and an electronic accelerometer. The water flow sensing tube uses the Adafruit ADXL345 electronic accelerometer to obtain the offset angles. Note that the offset angles are caused by the corresponding water velocity and flow direction. Meanwhile, the Arduino chip of the sensing tube sends the data of offset angles back to the Arduino chip deployed at the control box and then to the shore server for the regression estimation of water velocity. The regression estimation of water velocity will be discussed in detail in the next section.

## 4. Server-Side AI Programs

This section provides the details of the three server-side AI programs coded in Python and operated at the shore server. The shore server was built using the My-SQL database for data storage received from the offshore buoy. The data are water temperature, water velocity, and accelerometer’s deflection angles used to train a corresponding AI program individually. The results of the training provide water quality prediction. The following are further discussions of the three AI components.

### 4.1. Prediction for Water Temperature

This component is the first AI program with a prediction function for short-term water temperature. The following machine learning models were adopted, and we provided the methods for the implementation, and experiment results were generated.

(1)
*Long Short-Term Memory (LSTM)*


Short-term water temperatures do not change very often and are thus considered a short-term time series with mutual correlation features. Studies have shown that long short-term memory (LSTM) [41], a machine learning model, is suitable for time-series-related problems. It uses the mechanisms of the hidden layers to retrieve information from the correlations between time series. Figure 5 shows the multilayer structure and the widely used cell structure for LSTM. LSTM is a variant form of gated recurrent neural network (RNN), and proposed by Hochreiter and Schmidhuber in 1997 [42]. In addition, LSTM can overcome the problem of gradient disappearance or explosion, which usually appears at the conventional RNN while using an excessive number of layers in the time dimension. In particular, due to the gated control architecture, LSTM can suitably tune the self-loop weights and dynamically regulate the accumulated time scale. Among the gates of the LSTM, the input gate determines whether the inputs can be imported into the memory cell. Further, the memory cell can be linearly self-looping, whose weight is controlled by the forgetting gate. As for the output, its on–off is determined by the output gate. The gate units as mentioned earlier are the sigmoid functions, denoted as σ. The overall computation of an LSTM cell can be expressed as follows,
(1)$$\begin{array}{ccc} i_{t} & = & \sigma (W_{i}h+b_{i}),\\ f_{t} & = & \sigma (W_{f}h+b_{f}),\\ o_{t} & = & \sigma (W_{o}h+b_{o}),\\ c_{t} & = & tanh(W_{c}h+b_{c}),\\ m_{t} & = & f_{t}m_{t-1}+i_{t}c_{t},\\ h_{t} & = & tanh(o_{t}m_{t}) \end{array}$$
where $i_{t},f_{t},o_{t}$ denote the values of input gate, forget gate, and output gates, respectively; $W_{i},W_{f},W_{o}$ represent their corresponding weight matrices and $b_{i},b_{f},b_{o}$ are the corresponding bias vectors; tanh(·) is the hyperbolic tangent function; $h={\left[h_{t-1}^{T}\;x_{t}\right]}^{T}$ is the new hidden layer vector, $x_{t}$ is the input at time *t*, and (·) denotes the transpose operation; and $c_{t}$ and $m_{t}$ denote the cell’s new state vector and new memory vector, respectively. It is worth noting that the key mechanism of LSTM is the memory cell, which will remember the last input data of a time interval, making it suitable for solving the time-series problems such as the predicting water temperature and velocity. 

(2)
*Prediction Results of Water Temperature*


To predict water temperature, we designed an AI program to construct an LSTM-based network. We incorporated one of the hidden layers with 16 neurons in this network. Furthermore, we adopted the LSTM package of TensorFlow and the adaptive moment estimation (Adam) optimizer for the network. Further, the network is trained using 100 epochs and a dataset size of 734,000 water temperature data stored at the shore server. On the other hand, we use the other 314,573 water temperature data for testing the network. Given the space limitation of the figures, only 1700 tested results were shown in Figure 6, including the actual water temperatures and the prediction results of the network. For the mean-squared error (MSE), the result shows an in-between value of 0.021 °C. Most fish species are not sensitive to slight changes in water temperature, therefore, based on the results of Figure 6, the water temperature AI prediction program is a suitable method to implement the practical requirements in an offshore aquaculture environment.

### 4.2. Prediction Results for Water Velocity

Another AI function was constructed using an LSTM-based AI network to predict water velocity. Instead of a single hidden layer, two hidden layers were integrated for this other set of LSTM network since changes in the water velocity are relatively faster than that of water temperature. Furthermore, the number of neurons for the first and second layers are 96 and 64, respectively. We trained the new network using 100 epochs with a dataset of 36,237 water velocity data taken from the shore server. The other 15,531 water velocity data are used to test the network, and the prediction results are presented in Figure 7. Since the space for the figures is limited, only 900 results were captured in Figure 7. The result for MSE using the actual and the prediction value is 0.92 cm/s, which again implies that the AI program for water velocity prediction is suitable for offshore aquaculture.

### 4.3. Nonlinear Regression for Water Velocity Versus Accelerometer’S Depletion Angles

Instead of using an expensive flow meter for measuring current water velocity, we proposed a low-cost mechanism using a flow sensor tube. Figure 2 shows that the flow tube is hung under the steel skeleton of the offshore buoy. Furthermore, an Arduino Mega 2560 board and an electronic accelerometer are installed in the proposed flow tube. To measure water velocity, the Arduino Mega 2560 board first receives a command message from the control box and then instructs the accelerometer to measure the tube’s offset angles in the seawater. Then, the Arduino Meaga 2560 board sends the offset angle data back to the control box. Lastly, the data will be combined with other data such as water temperature and will be forwarded to the shore server for storage.

This offset angle data stored in the server will serve as the dataset to train a new AI program to perform the regression method for water velocity. We adopted the Scikit-learn package of TensorFlow to train the AI model using 560 offset angle data. The other 240 offset angle data were used to assess the feasibility of the flow tube and the results are shown in Figure 8. Furthermore, the MSE result of the regression is about 6.18 cm/s. On the other hand, the cost of the flow sensor tube is under USD 100. Therefore, considering the low-cost requirement of using the flow sensor tube and the MSE performance of the AI program, our proposed flow sensor tube is a viable and promising tool for offshore aquaculture.

## 5. Implementation Results and Discussions

To ensure that our proposed offshore buoy works in a real aquaculture environment, we deployed it at the offshore cages at Haikou Port, Pingtung, Taiwan, at a 2 km distance from the shore. The proposed buoy was in a fixed position near and outside the four cages to ensure that the ocean waves did not affect the devices that could lead to its dismantling since the flow sensor tube is just hung under the buoy. We set the distance between our proposed buoy and the nearest cage to about 15 m. On the other hand, the My-SQL server on the shore was deployed at the National Taiwan Ocean University. During the data collection, 734,000 water temperature and 36,237 velocity data were collected in the server and further utilized to train AI models to perform water quality prediction functions. The prediction results are shown in Figure 6 and Figure 7, respectively. As an additional feature, a mobile application (APP) was also integrated to provide a monitoring interface for aquaculture staff. Some of the APP interfaces and prediction results are shown in Figure 9.

In Figure 9a, we presented the system’s capacity to directly provide real-time water quality information from the offshore cage location. Furthermore, to offer comprehensive content, we integrated a web crawler in the APP to fetch local weather data from Taiwan Central Weather Bureau to provide additional valuable information to the aquaculture staff. With the integration of AI functions (discussed in Section 4), the short-term prediction information for water temperature and velocity are plotted at the top and bottom parts of Figure 9b.

In terms of costs, the main electronic equipment (including Arduino, accelerometer, GPS module, LoRa, battery, etc.), steel skeleton, control box, and two lifebuoys cost only about USD 1,469, 316, 150, and 79, respectively. Therefore, the total hardware cost for the proposed buoy system is approximately USD 2015, which could be attributed as practical and suitable for adoption in offshore aquaculture.

## 6. Conclusions

This paper focused on designing and implementing a low-cost, easy-to-build AI buoy system. The proposed buoy system autonomously measures temperature, velocity, DO, and salinity as water quality parameters and then forwards the collected data to the shore server for storage using a wireless communication channel. The water quality data provides aquaculture staff with real-time information and with prediction capabilities providing short-term information on water quality. To deliver a low-cost system, we combined low-priced electronic devices and AI functions to complete the architecture of our proposed buoy system to facilitate water velocity measurement.

The proposed buoy system for collecting water temperature and velocity measurement data facilitated the inclusion of prediction capabilities. The prediction results were further utilized to develop another AI function to estimate the feeding requirement for offshore cages using water velocity. This added knowledge of feeding amounts can assist the aquaculture staff in determining the suitable amounts of bait. This approach can lessen bait waste which lowers feeding costs and reduces ocean pollution.

## 7. Patents

The proposed buoy system has been applied as a patent in Taiwan, whose information is “ARTIFICIAL INTELLIGENCE BUOY, No. M-625369”.

## Figures and Tables

**Figure 1 sensors-22-04078-f001:**
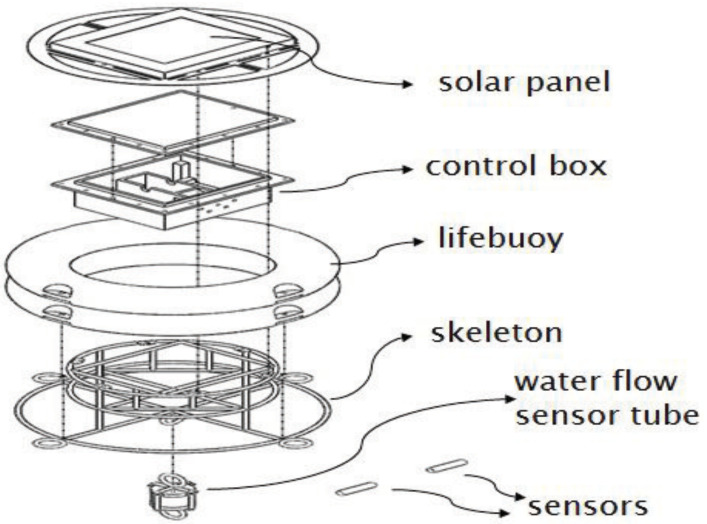
Architecture of the proposed offshore buoy.

**Figure 2 sensors-22-04078-f002:**
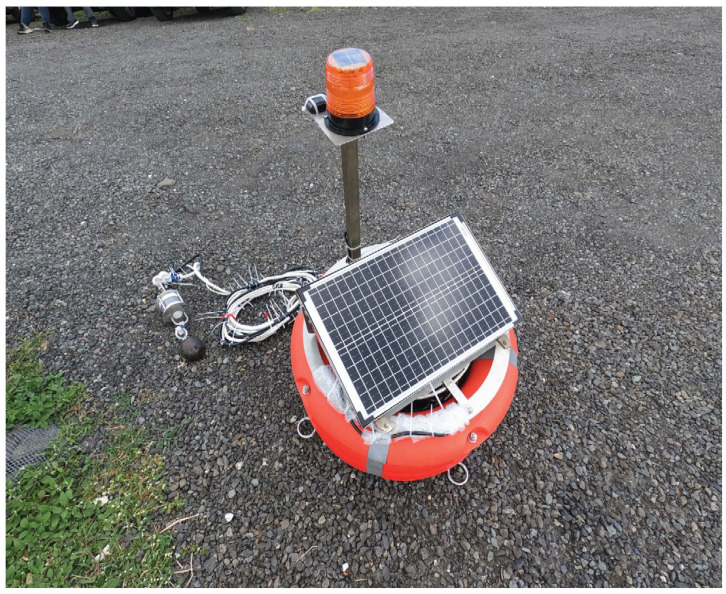
Proposed offshore buoy: main body and flow sensor tube.

**Figure 3 sensors-22-04078-f003:**
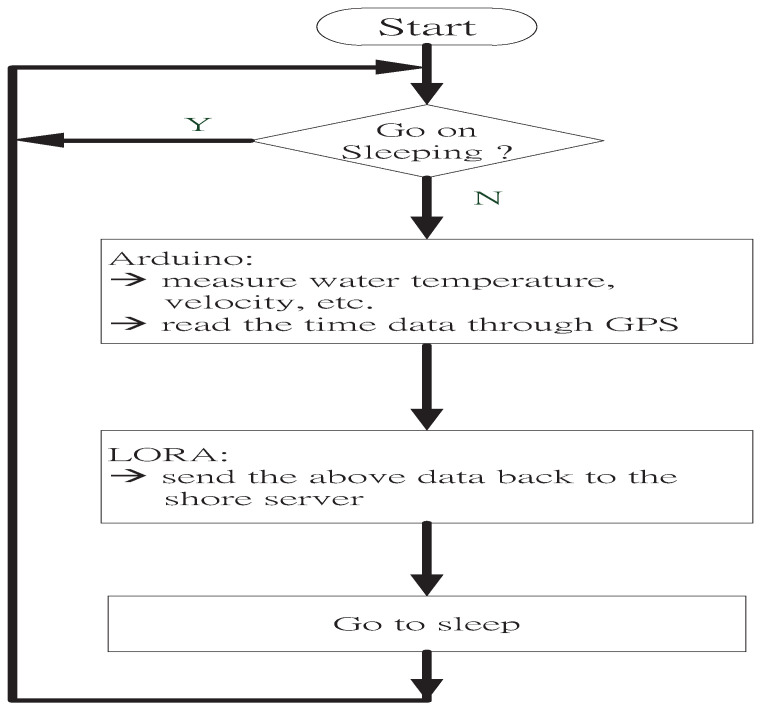
Operation flow of the proposed buoy.

**Figure 4 sensors-22-04078-f004:**
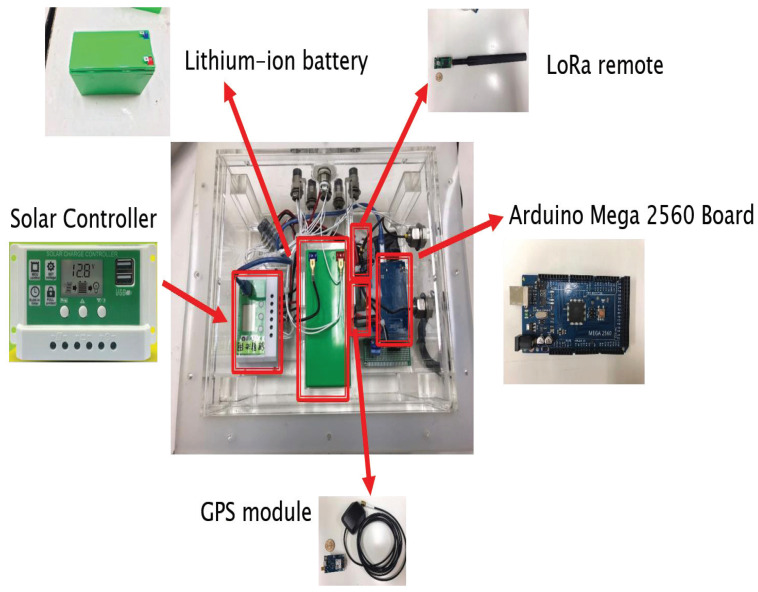
Control box of the proposed AI buoy.

**Figure 5 sensors-22-04078-f005:**
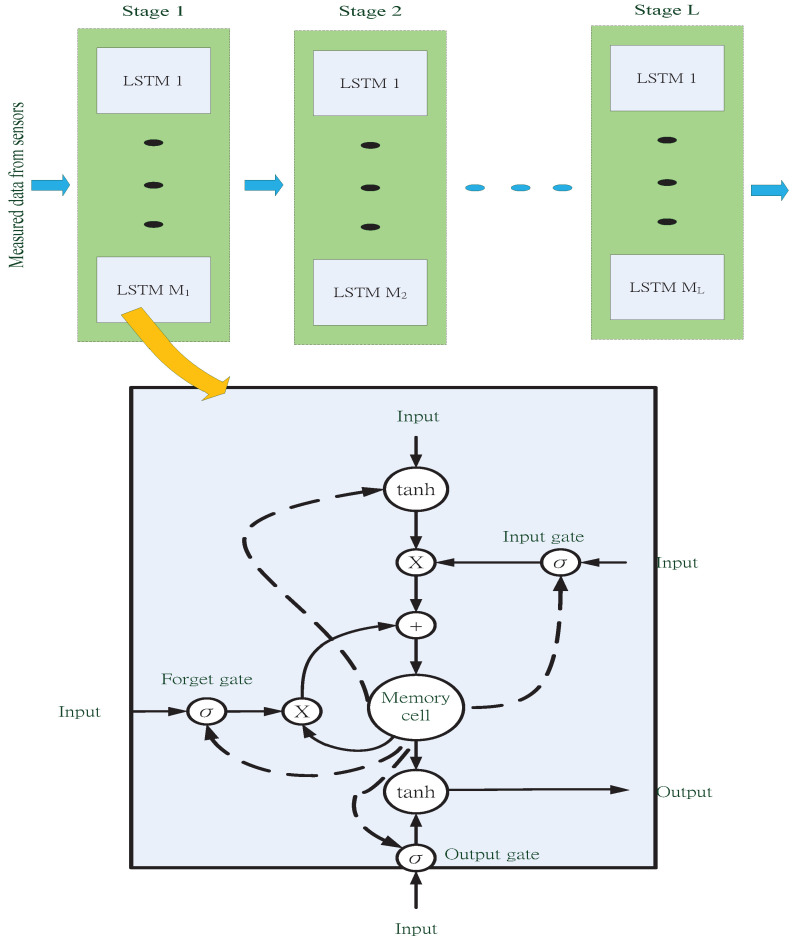
The multilaryer architecture and cell structure of long short-term memory (LSTM).

**Figure 6 sensors-22-04078-f006:**
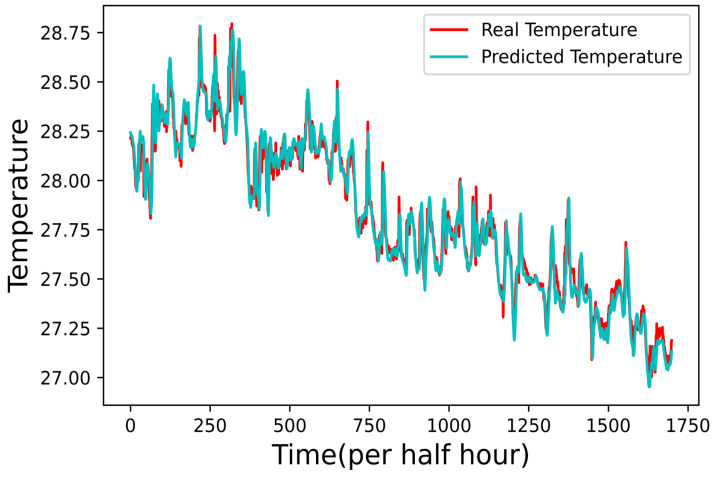
Prediction results of water temperature.

**Figure 7 sensors-22-04078-f007:**
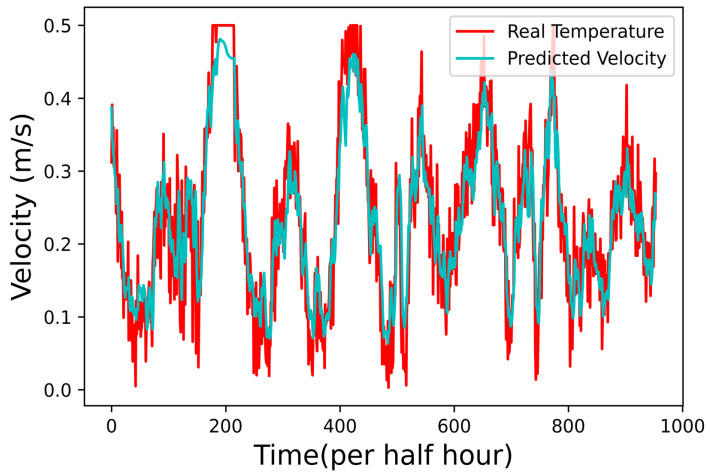
Prediction results of water velocity.

**Figure 8 sensors-22-04078-f008:**
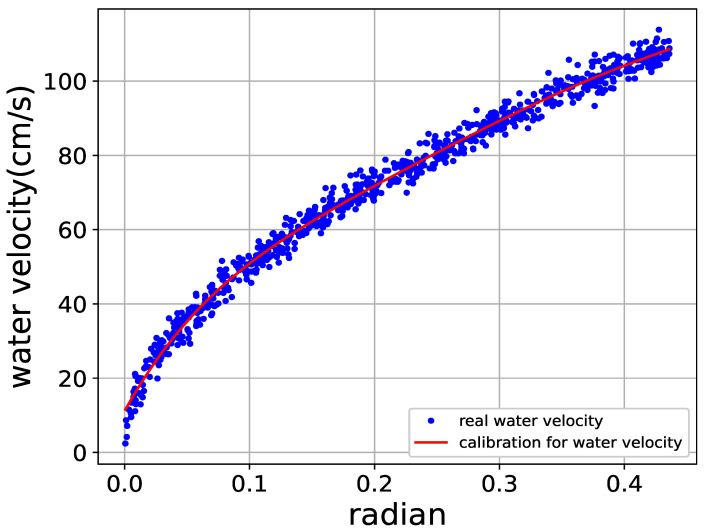
Regression of water velocity.

**Figure 9 sensors-22-04078-f009:**
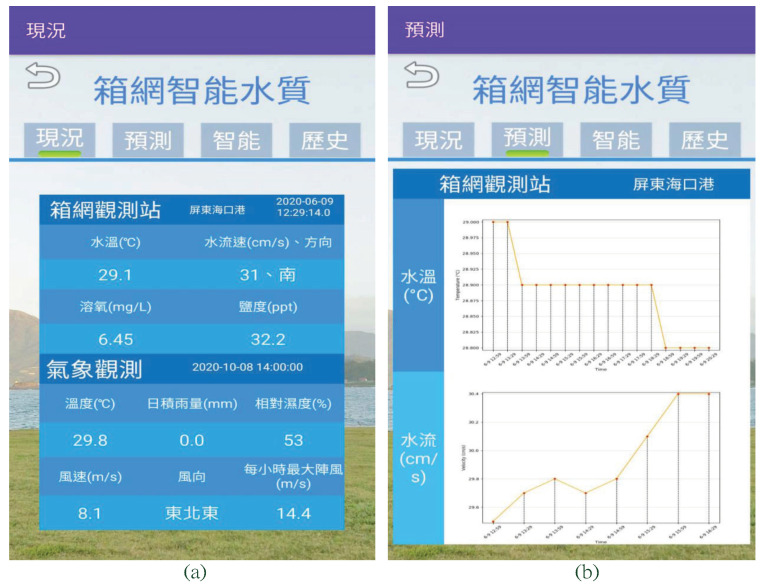
Smart water quality prediction, (**a**) current information, cage monitoring (at the top half part): water temperature, water velocity, dissolved oxygen, and salinity (from left to right and top to bottom), and weather information (at the lower half part): temperature, daily rainfall, humidity, wind speed, wind direction, maximum showers per hour (from left to right and top to bottom), and (**b**) prediction information, water temperature (at the top half part), and water velocity (at the lower half part).

**Table 1 sensors-22-04078-t001:** Technologies of [11,12,13,14,15,16,17,18].

Reference Number	Technologies
[11]	LoRa
[12]	CDMA, Zigbee
[13]	Machine vision
[14]	$I^{2}$C, GSM, Wi-Fi
[15]	IoT
[16]	LoRa
[17]	PLC, NB-IoT
[18]	GSM, Wi-Fi

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
