# Peer review of "A Low-Cost AI Buoy System for Monitoring Water Quality at Offshore Aquaculture Cages"

_sensors, 2022, doi:10.3390/s22114078_

Round 1
Reviewer 1 Report
General overview:
This paper is focused on some key problems, related to the development of a low-cost ai buoy system for monitoring real-time water quality at offshore cages. In fact, water quality is the key factor determining whether water quality is successful. The aim of the presented research was to assess the possibilities of assessing server-side AI programs to implement machine learning to further provide predictions of the short-term water quality.
Specific comments and some issues:
In this paper, a low-cost as well as easy-to-build AI buoy system is presented. Please try to respond to the following issue:
1. Wireless communications and the Internet of Things (IoT) are important. Please add more citations to the Introduction section.
2. Data choice can be verified by using real-time monitoring. Thanks to its use the significance level of particular characteristics can be evaluated, and it may be verified if all selected variables are essential in the learning process. How to apply your result in the practice?
3. This study evaluates methods used for offshore aquaculture. The aim is to identify an optimal combination of shape factors to measure. What is the application of your study for the high incorporation of a low-cost buoy system with artificial intelligence (AI)?
4. The scale of scrutiny of the measurement determines whether or not a temperature changes over time. Combining multiple long-term temperature changes and assessing the stresses to promote wireless communication modules would be helpful. What do you think?
5. Effects of nutrient and light limitation are observed. However, further research is required to determine the performance of the developed method for server-side AI programs, to enhance the stability. Water quality of the aquaculture pond, new production and net community production in the open ocean is currently a top priority due to more intense and more frequent changes. Do you agree with that opinion?
6. With the aim of elaborating an environmental application, accessible to anyone and with educational purposes, you have presented a method for more than two classes, or have been tested online. This article presents a novel method for the classification of quality and value. What do you suggest to improve your results?
7. You have presented the results, both in terms of segmentation and classification, considering a database of and implementation of water quality data. Which Figure helps to understand short-term prediction information of water temperature and velocity?
8. Figure 5 should be improved by adding descriptions for the sensors to measure the water quality such as water temperature and velocity.
Constructive feedback on paper:
This paper is focused on some key problems, related to the design of a feeding system for cage aquaculture based on IoT and AI Technology. The laboratory tests which are the basis for this paper were connected with research on the effectiveness of multi-parameter water quality monitoring devices for aquaculture. What kind of model can be applied to that problem?
The procedure for the classification comprises a real-time remote monitoring system for aquaculture water quality. Can you suggest something interesting bearing in the mind their impacts on the ecosystem and contaminants?
Over the years, the application of analytical chemistry in the industry has done much to improve the level and consistency of water pollution by agriculture. The status and quality of analysis of heavy metal levels in soil, tomatoes and selected vegetables from environmental and socio-economic impacts of mining should be elaborated. status of selected heavy metals dispersion from topsoil in and around automobile municipality. Heavy metal pollution or Potentially toxic elements (PTEs) like cadmium, lead and arsenic content is the source of pollution. Nowadays, instrumental analysis, particularly chromatography and spectroscopy, is applied to all aspects of the distribution of trace elements in the ecosystem, from heavy metals in an environment to ecological risk assessment for eutrophication and heavy metal pollution (mercury and lead contamination in species and sediments from lake). Key sensory parameters or components are determined directly – or indirectly using rapid methods such as robotic systems for automation of water quality monitoring and feeding in the aquaculture shade house and chemometric/statistical analysis. The structures of many complex sensory components have been elucidated by techniques such as mass spectrometry (MS) and nuclear magnetic resonance spectroscopy, the former usually being combined with gas chromatography (GC) or high-performance liquid chromatography (HPLC).
Over the past few decades, sample preparation techniques, especially extractive/focusing methods, have greatly facilitated instrumental analysis of possible risks for human consumption. It is interesting to consider which levels of heavy metals in wastewater or soil samples from open drainage channels can be evaluated during farming and mathematical modelling. What are the key concepts of research on the counting algorithm of residual feeds in aquaculture based on machine vision? Add something about future prospects of marine aquaculture.
This paper shows a careful examination and discussion of the measurement variability of aquaculture. This is a well-elaborated topic. However, any aspect of sustainable aquaculture development is not discussed. Assessment of cloud computing, Internet of things and artificial intelligence topics is not correctly reviewed. A remote supervision system for aquaculture platforms and identification of spatial dependence must be added. Please try to review:
• Tracking marine pollution;
• A flexible logging platform for long-term monitoring in marine or harsh environments;
• Assessment of edge artificial intelligence for the industrial internet of things applications as an industrial edge intelligence solution;
• Buoyant tethered sphere for marine current estimation.
Summary of the paper:
The authors are presenting an innovative perspective for sustainable aquaculture practice for remote monitoring systems based on ocean sensor networks for offshore aquaculture. The manuscript fits in the scope of the journal and reaches the quality the journal requires although still too long. The manuscript's conclusions offer a wide range of interesting proposals for policymakers and regulators. There is knowledge transfer that needs to be transmitted or communicated but it is not clear to whom (which government, farmers, educators). Nevertheless, there are some moderate changes the authors should consider in order to improve the understanding and reading of this research paper.
Author Response
Dear Reviewers,
The authors express their gratitude to the anonymous reviewers for their thorough reviews and many thoughtful comments and suggestions, which have enhanced the readability and quality of the manuscript. The modifications in this revision are detailed here. In addition, numerous grammatical and typographical errors have been corrected. The detail responses for the comments have been shown in the attached PDF file.
Best,
Hoang-Yang Lu, Chih-Yung Cheng, Shyi-Chyi Cheng, Yu-Hao Cheng, Wen-Chen Lo, Wei-Lin Jiang, Fan-Hua Nan, and Shun-Hsyun Chang

Reviewer 2 Report
Main comments
The subject of the article is interesting, but the manuscript must be revised to clarify the need for previsions based on AI tools. Offshore water velocity, direction, temperature, and other seawater parameters are a complex result of weather, air and water temperature, humidity, pressure, and also water composition. Is unclear the significance of previsions based only on past data, particularly if based on short time spam. In my opinion, the results should be validated against real data extraction from offshore farms. It is unclear in the text how the real data on temperature and water velocity were obtained.
I also recommend the revision of the article title. If the prevision feature is relevant, “real-time” must be removed. On the other hand, “aquaculture cages” may be clearer than only “cages”.
Minor comments
I recommend adding the main results to the abstract.
I recommend replacing words such as “lots of countries”, “related groups”, and “rich studies” with more objective statements.
I recommend a brief list of the technologies presented in works 11-18 (in line 23).
The work of Luna et al. [19] is not enough to support the statement in lines 24-25.
Using etc. is not recommended in line 28 because the authors intend to give examples of the most relevant water quality parameters.
Please add a reference to support the statement in lines 30-31. Same recommendation for support of the statement in lines 35-37.
Please check the need for “back” in line 42.
I recommend writing “In the work of Marchant et al. [24], …” in line 54. In addition, write “Further, Beddows et al. [25] …” in line 60, …
It is unclear the need for references [27] and [28] in line 67.
I recommend moving Figures 1 and 2, and the description of the set-up to section 2. Section 1 must include a shorter description of the work, mainly focusing on its novelty compared to other works. In addition, the need for using DO and salinity sensors must be presented.
Is the buoy fixed inside/outside the cages?
What is the size of the covered area? How the sensors read at buoy location can be representative of an entire aquaculture farm?
Why the water velocity at present time is predicted? In addition, the applied AI tools should be described in detail.
Why the need for a GPS module?
I recommend giving the costs in an international currency such as USD or EUR. In addition, a comparison with commercial-available equipment should be presented to allow the classification as “low-cost” solution.
The manufacturers and the models of all equipment must be presented in the text to alloy a clear evaluation of the work carried out and allow for future replication. References 30 to 34 must be removed.
References 35, 36, and 39 can be removed.
Author Response
Dear Reviewers,
The authors express their gratitude to the anonymous reviewers for their thorough reviews and many thoughtful comments and suggestions, which have enhanced the readability and quality of the manuscript. The modifications in this revision are detailed here. In addition, numerous grammatical and typographical errors have been corrected.
Hoang-Yang Lu, Chih-Yung Cheng, Shyi-Chyi Cheng, Yu-Hao Cheng, Wen-Chen Lo, Wei-Lin Jiang, Fan-Hua Nan, and Shun-Hsyun Chang

Reviewer 3 Report
The purpose and main objective of the article are not disclosed. The authors offer an inexpensive, as well as an easy-to-assemble buoy system with artificial intelligence. There is no comparison in the text with fights with similar characteristics. In addition, the temperature and speed of water movement are selected as criteria for water quality. For what? This is a set of big data values for machine learning. But what's next, how does it affect the market of these devices or what new has the proposed design allowed to achieve? Unfortunately, I have to admit that the scientific value of the article is not obvious.
Author Response
Dear Reviewers,
The authors express their gratitude to the anonymous reviewers for their thorough reviews and many thoughtful comments and suggestions, which have enhanced the readability and quality of the manuscript. The modifications in this revision are detailed here. In addition, numerous grammatical and typographical errors have been corrected.
Best,
Hoang-Yang Lu, Chih-Yung Cheng, Shyi-Chyi Cheng, Yu-Hao Cheng, Wen-Chen Lo, Wei-Lin Jiang, Fan-Hua Nan, and Shun-Hsyun Chang

Round 2
Reviewer 1 Report
The manuscript has been enhanced in several ways and the missing information has been added. The main issues I raised have been addressed. There are, however, a few minor issues I would like addressed before it can be accepted. Editing the paper and improving the grammar and syntax would be helpful.
Thank you for your thoughtful comments. The results are as expected.
I think the target has improved significantly. The work methodology appears to have improved.
The conclusions have been reviewed and improved following an appeal to my remark.
Author Response
The authors express their gratitude to the anonymous reviewers for their thorough reviews and many thoughtful comments and suggestions, which have enhanced the readability and quality of the manuscript. The modifications in this revision are detailed here. In addition, numerous grammatical and typographical errors have been corrected.

Reviewer 2 Report
I aknowledge the efforts of the authors to answer all questions and to improve the article.
Author Response

(The authors gave the same response as above.)

Reviewer 3 Report
The responses to the comments are satisfactory.
Author Response

(The authors gave the same response as above.)
